# Radioiodine in Differentiated Thyroid Carcinoma: Do We Need Diagnostic Pre-Ablation Iodine-123 Scintigraphy to Optimize Treatment?

**DOI:** 10.3390/diagnostics11030553

**Published:** 2021-03-19

**Authors:** Elizabeth J. de Koster, Taban Sulaiman, Jaap F. Hamming, Abbey Schepers, Marieke Snel, Floris H. P. van Velden, Lioe-Fee de Geus-Oei, Dennis Vriens

**Affiliations:** 1Department of Radiology and Nuclear Medicine, Radboud University Medical Center, 6525 GA Nijmegen, The Netherlands; 2Department of Radiology, Section of Nuclear Medicine, Leiden University Medical Centre, 2333 ZA Leiden, The Netherlands; taban.sulaiman@gmail.com (T.S.); F.H.P.van_velden@lumc.nl (F.H.P.v.V.); L.F.de_Geus-Oei@lumc.nl (L.-F.d.G.-O.); D.Vriens@lumc.nl (D.V.); 3Department of Surgery, Leiden University Medical Centre, 2333 ZA Leiden, The Netherlands; J.F.Hamming@lumc.nl (J.F.H.); A.Schepers@lumc.nl (A.S.); 4Department of Endocrinology, Leiden University Medical Centre, 2333 ZA Leiden, The Netherlands; M.Snel@lumc.nl

**Keywords:** differentiated thyroid carcinoma, radioiodine ablation, I-131, thyroid remnant, lymph node metastasis, posttherapy scintigraphy, pre-ablation diagnostic scintigraphy, I-123

## Abstract

Changing insights regarding radioiodine (I-131) administration in differentiated thyroid carcinoma (DTC) stir up discussions on the utility of pre-ablation diagnostic scintigraphy (DxWBS). Our retrospective study qualitatively and semi-quantitatively assessed posttherapy I-131 whole-body scintigraphy (TxWBS) data for thyroid remnant size and metastasis. Findings were associated with initial treatment success after nine months, as well as clinical, histopathological, and surgical parameters. Possible management changes were addressed. A thyroid remnant was reported in 89 of 97 (92%) patients, suspicion of lymph node metastasis in 26 (27%) and distant metastasis in 6 (6%). Surgery with oncological intent and surgery by two dedicated thyroid surgeons were independently associated with a smaller remnant. Surgery at a community hospital, aggressive tumor histopathology, histopathological lymph node metastasis (pN1) and suspicion of new lymph node metastasis on TxWBS were independently associated with an unsuccessful treatment. Thyroid remnant size was unrelated to treatment success. All 13 pN1 patients with suspected in situ lymph node metastases on TxWBS had an unsuccessful treatment, opposite 19/31 (61%) pN1 patients without (*p* = 0.009). Pre-ablative knowledge of these TxWBS findings had likely influenced management in 48 (50%) patients. Additional pre-ablative diagnostics could optimize patient-tailored I-131 administration. DxWBS should be considered, especially in patients with pN1 stage or suspected in situ lymph node metastasis. Dependent on local surgical expertise, DxWBS is not recommended to evaluate thyroid remnant size.

## 1. Introduction

Management of differentiated thyroid carcinoma (DTC) is becoming more patient-tailored. This includes ongoing discussions on the application of post-thyroidectomy radioactive iodine-131 (I-131), previously one of the treatment mainstays for most DTC. Post-thyroidectomy I-131 serves three main goals: ablation of any thyroid remnant to facilitate accurate follow-up, adjuvant therapy for the risk of any remaining (microscopic) disease in the thyroid bed or cervical lymph nodes, and therapy for known locoregional or distant residual disease [1,2,3]. Although its anatomical substrate is often negligible, the biochemical presence of a minimal remainder of thyroid tissue could interfere with accurate follow-up, which is based on the high specificity of serum thyroglobulin (Tg) and cervical ultrasound [4]. A larger thyroid remnant is generally found following two-stage surgery (i.e., diagnostic lobectomy followed by completion thyroidectomy) or surgery performed by less experienced surgeons [5,6].

The standard posttherapy whole-body scintigraphy (TxWBS) is a helpful ancillary for postoperative tumor staging. For a more timely risk stratification and fitted I-131 dosimetry, pre-ablation diagnostic whole-body scintigraphy (DxWBS) can be performed using a low dose of I-131, I-123, or I-124 [7,8]. It is believed that DxWBS and the consequent individual treatment optimization could ultimately reduce morbidity, tumor dedifferentiation, and persistent or recurrent disease. Possible management alterations include alterations to the initial I-131 dose following the detection of previously unknown in situ lymph node metastases or distant metastases [4,9,10]. Or, in case of a large thyroid remnant, two-stage I-131 treatment may be preferred: an initial low I-131 dose ablates the benign thyroid remnant, followed by a higher adjuvant dose treating any residual and less iodine-sensitive malignant cells. Two-stage I-131 treatment prevents extended hospitalization and adverse deterministic effects on surrounding tissues caused by higher and prolonged I-131 retention in the neck, at a minor risk of thyroid stunning [11,12]. Third, surgical re-exploration prior to I-131 treatment could be considered in very large thyroid remnants or lymph node metastases larger than 10 mm, as I-131 alone unlikely suffices in these cases and likely increases morbidity [13].

Currently, DxWBS is not common practice due to its assumed limited impact on patient management in light of high-quality surgery [14,15]. The Dutch national guidelines suggest that it should be a local consideration, as the yield of DxWBS depends on case-mix, extent of preoperative diagnostic imaging, and surgical expertise [16]. Yet, the ongoing global debate to limit the exposure to radioiodine also stirs up the discussion on DxWBS for optimization of I-131 therapy [14,17,18,19,20]. Despite reservations about the sensitivity of DxWBS, management changes are reported in 25–53% of patients following clinically significant findings on I-123 DxWBS [8,9,21,22,23,24,25].

At our tertiary care center, DxWBS is currently only performed selectively if a large thyroid remnant or undiagnosed lymph node metastasis are surmised, such as in patients with a deliberate irradical thyroidectomy or patients who had their thyroid surgery in a low-volume community hospital.

The current study aimed to investigate the need for pre-ablation I-123 DxWBS at our hospital. The primary objective was to evaluate TxWBS images for clinically relevant findings that had likely influenced the course of treatment, had they been known prior to I-131 ablative therapy. Secondarily, associations between TxWBS findings, the size of the thyroid remnant, treatment success and various clinical, surgical, and histopathological factors were assessed.

## 2. Materials and Methods

### 2.1. Patient Selection

This retrospective study included all consecutive DTC patients referred to our tertiary care center at any time in their disease course but prior to their first I-131 administration between December 2010 and July 2015. Patients were excluded if distant metastases were present at the time of diagnosis, if pre-ablation DxWBS, surgical re-exploration or radiotherapy was performed, if TxWBS was not performed on our standard-of-care single-photon emission computed tomography (SPECT) scanner, or if a formal follow-up moment was unavailable approximately nine (range 6–12) months after I-131 administration. Ethical study review was waived by the institutional review board. No informed consent was required.

### 2.2. Patient Management

At our institution, DTC is diagnosed and treated in accordance with Dutch national guidelines and in close multidisciplinary collaboration of endocrinologists, endocrine surgeons, pathologists, and nuclear medicine physicians [14,16]. Approximately 30 new patients are surgically treated for DTC each year by dedicated endocrine surgeons: surgeon A (approximately 3 years of experience at the start of patient inclusion), surgeon B (30 years of experience), and surgeon C (24 years of experience). Other surgeons perform a mere limited number of procedures. Histopathology reports comply with the current World Health Organization guidelines [26].

Postoperatively, all patients included in this study underwent one or more cycles of I-131 treatment. Following Dutch national guidelines, I-131 dosing is based on pTNM-status (TNM staging system, seventh edition [27]), histologic tumor origin, resection margins and extracapsular growth of lymph node metastases (Appendix A). Like most EU-countries, these guidelines are more liberal regarding I-131 administration than the current ATA guidelines [14,16]. Patients were prepared by either (at least 4 weeks of) thyroid hormone withdrawal or recombinant thyroid stimulating hormone (rhTSH), resulting in serum thyroid stimulating hormone (TSH) levels ≥30 mEh/L. Stimulated serum thyroglobulin (sTg) levels were measured on the day of I-131 administration or the fifth day after the first rhTSH administration. TxWBS was performed one week (7 ± 1 days) after I-131 administration. Approximately nine months after I-131 administration (6–12 months in low-risk and 6–9 months in intermediate- and high-risk patients), patients were first evaluated with sTg and ultrasound, and I-131 DxWBS upon indication. Successful treatment was defined as a complete response to treatment at the nine-month evaluation: no clinical, biochemical (i.e., sTg < 0.5 ng/mL in absence of Tg-antibodies), or structural (i.e., on imaging) evidence of disease. Unsuccessful treatment was defined as an incomplete or indeterminate response to treatment: suspicion or evidence of persistent or progressive disease, either by detectable Tg levels (assay detection limit >0.5 ng/mL), presence of Tg antibodies, or on imaging studies (Appendix A) [14,28].

### 2.3. TxWBS

TxWBS imaging was performed on a Symbia T6 SPECT/CT scanner (Siemens Healthcare, Erlangen, Germany) equipped with parallel-hole, high-energy collimators, using a 15% energy window set at 364 keV. The scanning protocol included anterior and posterior whole-body planar imaging (15 cm/minute, a 256 × 256 matrix, no zoom), as well as SPECT/CT of the neck from the parotid glands until the inferior lung lobes (45 projections, 25 s per projection, automatic body-contour, 128 × 128 matrix, no zoom). When suspicious uptake was observed elsewhere during planar imaging, the SPECT/CT was extended caudally on discretion of a nuclear medicine physician. A CT image was acquired (130 kV, 40 mAs, 2-mm collimation and 0.8 pitch) for attenuation correction purposes. SPECT images were reconstructed using an iterative reconstruction algorithm (Flash 3D; four iterations, two subsets) with attenuation correction.

### 2.4. TxWBS Qualitative Assessment

Two experienced nuclear medicine physicians (L.G. and D.V., 23 and 9 years of experience, respectively) performed an independent and blinded visual assessment of all TxWBS planar and SPECT/CT images. A consensus meeting resolved discordant reviews. The reviewers scored any non-physiological uptake in the neck (defined as the region between mandibula and suprasternal notch) according to its anatomical location. Any I-131 uptake within the thyroid bed was attributed to remaining benign or malignant thyroid tissue, and is further referred to as thyroid remnant. The size of the thyroid remnant was categorized as small, medium, or large. The thyroid bed was defined as the area from the upper plane of the hyoid to the base of the neck, extending laterally around the trachea on either side of the midline. Three levels were distinguished according to their prior anatomical structures (Figure 1). The boundary between levels A and B was set at the cornu inferior of the thyroid cartilage, cranial to where the recurrent laryngeal nerve enters the cricothyroid membrane. Directly below on either side of the cricoid cartilage, Berry’s ligament is found, known for its proximity to the recurrent laryngeal nerve. The transition from cricoid to trachea indicated the boundary between levels B and C. Non-physiological uptake within the neck but outside the thyroid bed was ascribed to suspected cervical lymph node metastases, with specification of levels I–VII. Finally, any non-physiological uptake outside the neck was assessed, including suspected distant metastases, ectopic thyroid tissue, and thymus. Findings that would likely have changed the patients’ treatment course if they had been detected prior to TxWBS, were defined clinically relevant.

### 2.5. TxWBS Semi-Quantitative Assessment

Blinded semi-quantitative analysis of I-131 uptake was performed using OsiriX™ (version 8.5 Lite; Pixmeo, Geneva, Switzerland). On SPECT/CT imaging, uptake counts in the thyroid bed were semi-automatically measured using an isocontour-based volume- of-interest (VOI). Any physiological or metastatic uptake was manually excluded from the VOI. Second, background uptake (b) was measured. Multiple background locations were assessed for eligibility (Appendix A). Consequently, a predefined fixed-volume spherical background VOI (ø 20 mm) in the shoulder was used (Figure 2) [29,30]. Next, the size of the thyroid remnant was semi-quantitatively assessed by calculating a background- and volume-corrected (mm^3^ for SPECT/CT) value for the relative additional iodine uptake in the thyroid bed (thy) above background (b), further referred to as thyroid-remnant-to-background ratio (TRB ratio). This allowed correction for inter-patient variability, including intensity of iodine uptake for different I-131 doses and time-specific variances. It was calculated using the formula
(1)$$TRB\;ratio\;=\;\frac{{\left[counts\right]}_{thy}-\frac{{\left[counts\right]}_{b}}{{\left[volume\right]}_{b}}{\times [volume]}_{thy}}{\frac{{\left[counts\right]}_{b}}{{\left[volume\right]}_{b}}{\times [volume]}_{thy}}\;=\;\frac{{\left[counts\right]}_{thy}}{{\left[counts\right]}_{b}}\times \frac{{\left[volume\right]}_{b}}{{\left[volume\right]}_{thy}}-1.$$

As such, a higher TRB ratio represents higher uptake in the remnant above the background and thus a larger thyroid remnant; a TRB ratio of 0 represents no iodine uptake.

### 2.6. Statistical Analysis

Statistical analysis was performed using SPSS software (version 23; IBM Corp, Armonk, NY, USA). Categorical variables were analyzed using chi-square or Fisher’s exact test. Continuous data were assessed for (log)normality using histograms and skewness/kurtosis metrics, and analyzed using either parametric tests (Student’s *t*-test) or their nonparametric test equivalents (Mann–Whitney U, Kruskal–Wallis one-way ANOVA, or Spearman’s correlation coefficient), where appropriate. Correlation coefficients were expressed as the fraction of explained variance by their value squared (R^2^). Univariable and multivariable prediction models were generated using the continuous TRB ratio (linear regression modelling) or the binary ‘treatment success’ (logistic regression modelling) as dependent variable, and known predictive factors (e.g., operator experience) and parameters univariably trending towards a predictive association (*p* < 0.1) as independent variables. Multivariable models were iteratively generated using the backward stepwise method, optimizing for likelihood ratio. A *p*-value < 0.05 was considered statistically significant.

## 3. Results

From December 2010 through July 2015, 157 patients who underwent thyroidectomy for DTC were identified, of whom 124 fulfilled our inclusion criteria. Of these, 27 were excluded: five had proven distant metastases at the time of diagnosis, two underwent pre-ablation DxWBS, one had radiotherapy, one underwent surgical re-exploration of the neck, 12 TxWBSs were not performed on the Symbia T6 SPECT/CT scanner, and six patients had their postoperative treatment at another hospital. Finally, 97 patients were included. Table 1 shows their baseline characteristics. I-131 was administered after a median of 7.9 weeks (range 2.6–23) following total thyroidectomy.

### 3.1. Findings on TxWBS

On TxWBS, a scintigraphic thyroid remnant was observed in 89 of 97 (92%) patients. Forty-one (42%) patients had I-131 uptake in one location; 48 (49%) in more than one location, with a maximum of five. Figure 1 shows the most frequent anatomical sites. Remnants were found ipsilateral to the primary tumor in 29 (30%), contralateral in 28 (29%) and bilateral in 32 (33%) patients. Moreover, TxWBS located 37 suspected lymph node metastases in 26 patients (27%), of which 13 (50%) patients were pN0/x. Suspected new lymph node metastases were found in 4 of 18 (22%) low, 3 of 38 (13%) intermediate, and 17 of 41 (42%) high ATA risk patients (*p* = 0.017, Fisher’s exact test). The observation of suspected lymph node metastases was unrelated to prior surgical lymph node dissection (*p* = 0.653, chi-square), TNM N-stage (*p* = 0.925, Fisher’s exact test), the surgeon (*p* = 0.654, Fisher’s exact test), and sTg or presence of anti-Tg antibodies at the time of I-131 treatment (*p* = 0.644 and *p* = 0.089, respectively, Fisher’s exact test). Unexpected distant metastases were diagnosed on TxWBS in six (6%) patients: two with osseous and three with pulmonary metastases, one with both. Three of these six patients also had new suspected lymph node metastases on TxWBS.

### 3.2. Thyroid Remnant Size

SPECT/CT images were available for semi-quantitative assessment in 87 of 97 patients (89%). The median TRB ratio, representing the size of the thyroid remnant, was 11.6 (interquartile range (IQR): 7.07–28.7). A larger thyroid remnant was found if the remnant was located at least at the former pyramidal lobe (multivariable linear regression, R2 = 0.079, *p* = 0.009). A significantly smaller thyroid remnant was seen in patients who underwent surgery for suspected malignancy (oncological intent) as opposed to a benign surgical intent (i.e., goiter surgery), initial total thyroidectomy as opposed to two-stage thyroidectomy, and any lymph node resection of one or more cervical levels (Table 2). The smallest remnant was seen if formal neck dissection of the central compartment (level 6/7) was performed. Most surgical procedures were performed by surgeon A (58 procedures), surgeon B (47 procedures), and surgeon C (20 procedures). Between them, thyroid remnant size did not differ. However, surgery performed by a duo of our dedicated thyroid surgeons A, B, or C resulted in a significantly smaller remnant than surgery by a single dedicated surgeon or any other duo of surgeons. In 13 of 25 (48%) patients who underwent two-stage surgery, both lobectomy and completion thyroidectomy were performed by the same surgeon; this was unrelated to thyroid remnant size.

Multivariable linear regression analysis demonstrated that surgery with oncological intent (standardized coefficient beta = −0.335, *p* = 0.002) and surgery by a duo of dedicated thyroid surgeons (beta = −0.246, *p* = 0.018) were significant independent predictors of a smaller thyroid remnant (R^2^ = 0.144, *p* = 0.001). Two-stage surgery, any lymph node dissection, individual operator experience and I-131 dose were not independent predictors of the size of the remnant.

### 3.3. Treatment Success

Nine months after initial I-131 therapy, treatment was successful in 41 of 97 (42%) patients (Appendix A). ROC curve analysis showed that at a cut-off TRB ratio of 24.4, thyroid remnant size had a most optimal 80.0% sensitivity and 59.5% specificity to predict treatment success (Appendix A). A large thyroid remnant was therefore defined as TRB ratio ≥24.4.

On univariate analysis, any lymph node dissection, at least one thyroid surgery at a referring community hospital as opposed to all surgeries at our tertiary care center, a difficult surgical procedure (i.e., due to high patient body mass index, tissue adhesions, scar tissue, an aberrant course of the recurrent laryngeal nerve, difficult access to a retrotracheal thyroid remnant or other anatomical variations), an irradical resection, higher TNM pT or pN stage, administered I-131 dose, a smaller thyroid remnant, presence of previously unknown distant metastases on TxWBS, or higher sTg at time of the I-131 administration were associated with an unsuccessful treatment (Table 3).

Multivariable logistic regression modelling showed that at least one surgery at a referring hospital, an aggressive tumor histopathology (i.e., diffuse sclerosing, tall cell or oncocytic variant of papillary thyroid carcinoma, Hürthle cell carcinoma, and poorly differentiated thyroid carcinoma), TNM stage N1b and suspected lymph node metastasis on TxWBS were independently associated with an unsuccessful treatment (chi-square 20.416, R^2^ = 0.281, *p* < 0.001) (Table 4). Surgical difficulties, extrathyroidal tumor extension, an irradical resection, thyroid remnant size, pT-stage, and serum sTg were not independently predictive of treatment success. Presence of new distant metastasis on TxWBS was not included in the model because it was a constant; I-131 dose was not included to avoid multicollinearity. The anatomical location of the thyroid remnant was not associated with the nine-month treatment success rate (multiple logistic regression, R^2^ = 0.000, *p* = 0.499).

### 3.4. Lymph Node Metastasis

Treatment was more frequently unsuccessful in the 44 patients with lymph node metastasis on histopathology (pN1a/b) than in the 38 without (pN0) or 15 with unknown lymph node status (pNx) (*p* = 0.030, Table 3). Of pN1 patients, all 13 (100%) patients in whom additional suspected lymph node metastases were seen on TxWBS had evidence of disease at the nine-month follow-up moment, opposite 19 of 31 (61%) pN1 patients with no additional metastases on TxWBS (*p* = 0.009). Treatment success was similar in pN0/x patients with or without lymph node metastases on TxWBS (6/13 (46%) versus 18/40 (45%) unsuccessful treatment, respectively, *p* = 0.942).

### 3.5. Consequences for Patient Management

One or multiple clinically relevant findings were seen on TxWBS in 48 of 97 patients (50%) (Appendix A). Adhering to the current Dutch national guidelines, the patient’s course of treatment would likely have changed if these findings had been diagnosed prior to I-131 therapy. In 27 (28%) patients with a large thyroid remnant (large on qualitative assessment of planar images (*n* = 2) or TRB ratio ≥24.4 (*n* = 25)), two-stage I-131 therapy or re-exploration surgery would have been considered. A minimum cumulative I-131 dose of 3.7 GBq would have been advised [14,16]. As the current study shows that thyroid remnant size is unrelated to treatment success, a higher I-131 dose may not be justified.

Management changes would have also been likely in 19 of 26 (53%) patients with suspected in situ lymph node metastasis on TxWBS. In patients with one (*n* = 11) or multiple (*n* = 4) small (diameter <10 mm on CT imaging) suspected lymph node metastases, a higher I-131 dose would have been advised: 5.55 GBq or 7.4 GBq, respectively [16]. In four patients with large (>10 mm) lymph node metastasis, either re-exploration surgery or 7.4 GBq of I-131 could have been considered. The latter four all had persistent or progressive disease after nine months follow-up.

In five of six (83%) patients with new distant metastasis, a higher 7.4 GBq I-131 dose would have been recommended.

## 4. Discussion

The current study aimed to investigate whether DTC patients at our hospital could benefit from pre-ablation I-123 DxWBS by assessing the TxWBS images of 97 patients for clinically relevant findings. TxWBS showed a large thyroid remnant in 27 (28%), lymph node metastasis in 26 (27%), and distant metastasis in six (6%) patients. These results are similar to previous studies reporting 35% new lymph node metastases and 8% distant metastasis [9,32]. Previous series reported 29–53% management changes following pre-ablative DxWBS [9,21]. In accordance with the Dutch national guidelines, pre-ablative knowledge of the TxWBS findings would likely have influenced the course of treatment in 48 of 97 (50%) patients in the current study. However, the observation of a large thyroid remnant should be clearly distinguished from suspected (lymph node) metastasis. Therapeutic and prognostic consequences of a large thyroid remnant are debatable. The current study showed that the size and anatomical location of a thyroid remnant were unrelated to the nine-month treatment success. Although pursuing initial adequate thyroid remnant ablation benefits Tg-based follow-up and multidisciplinary decisions on any subsequent I-131 administrations, management changes based on the (size of the) thyroid remnant seem unnecessary in the context of treatment success. Impact on the long-term oncological outcome cannot be deduced from the current study.

The inverse association between thyroid remnant size and treatment success on univariate analysis may be explained by several related factors: more complex cases were more frequently operated by a duo of dedicated thyroid surgeons, such as 16 of 25 (64%) patients with preoperatively proven lymph node metastasis compared to 19 of 72 (26.4%) patients without (*p* = 0.003). Surgery by a duo of dedicated thyroid surgeons and performance of any formal lymph node dissection were correlated with a smaller thyroid remnant. More complex cases, i.e., patients with lymph node metastasis (pN1 stage), more frequently had evidence of disease at the 9-month follow-up moment. On multivariate analysis, pN1b stage was the remaining factor that was independently related to treatment success.

An unexpected large thyroid remnant could have potential undesired consequences. Its high and prolonged I-131 uptake may lengthen hospital admission for I-131 ablative therapy and cause deterministic toxicity to skin and surrounding soft tissues. In the current study, thyroid remnant size was unrelated to the length of hospital stay if corrected for administered dose and patient age. Following the overall high-quality thyroid surgery at our institution, only four patients (4% of the total population) had such large thyroid remnants that this could have extended their hospital stay. Overall, dependent on local surgical expertise, we would not recommend routine DxWBS to merely evaluate the size of the thyroid remnant.

Pre-ablative diagnosis of previously unknown metastases is of greater relevance. Lymph node metastasis is a known major contributor to recurrent disease, disease-related morbidity, and adverse prognosis in DTC, even if I-131 treatment of lymph node metastasis initially seems successful in a majority of patients [3,33,34,35,36]. For these intermediate- and high-risk patients, prior knowledge of in situ lymph node metastasis could refine risk stratification and patient-tailored I-131 dosimetry. In our study, TxWBS showed suspected in situ lymph node metastasis in 26 of 97 (27%) patients. Although similar to previously published rates, this relatively high number suggests that preoperative staging ultrasound procedures may require further optimization before introduction of an additional diagnostic is considered [9,32]. Improving existing thyroid ultrasound procedures is likely less costly and laborious, but may equally benefit individual patient outcomes, depending on the current local expertise. However, in the current study, most iodine-positive lymph nodes were non-enlarged on (SPECT)CT. It is uncertain how many of these would have been detected on a more comprehensive preoperative ultrasound.

In the current study, suspected lymph node metastasis on TxWBS were confirmed on ultrasound and/or DxWBS in 6 (23%) and no longer visualized in 20 of 26 (77%) patients at the nine-month evaluation. Even so, this TxWBS finding was related to biochemical or structural evidence of persistent disease, in particular in patients that already had proven lymph node metastasis on histopathology (pN1). The majority (85%) of these patients were treated with 5.55 GBq or less. According to our local protocol, an initial higher 7.4 GBq dose had been administered if this in situ metastatic disease had been diagnosed prior to I-131 therapy. We believe that this had likely improved the chance of an immediate successful treatment, possibly omitting subsequent rounds of I-131. On population level, it likely lowers the administered cumulative dose [2,9].

TNM stage N1b and suspected lymph node metastasis on TxWBS were independently associated with evidence of disease at nine months. Subgroup analysis in pN0/x patients showed no correlation between suspected lymph node metastasis on TxWBS and unsuccessful treatment. Any existing association may be minor and limited by study power or presence of confounding variables. No recommendation on DxWBS for pN0/x patients can be made based on the current study.

The current study assumed that TxWBS findings would reflect DxWBS findings. As no actual DxWBS was performed, results from the current study are subject to a certain level of uncertainty and speculation. The actual benefits of DxWBS may be more limited. Discordance between I-123 DxWBS and I-131 TxWBS was previously reported by one of our co-authors [23,24]. The relatively short half-life of I-123, limited radioiodine uptake in less-oxygenated metastatic tissues (especially osseous metastasis) and varying scanning protocols (i.e., acquisition time) could contribute to insufficient target-to-background contrast and limited diagnostic capability of DxWBS [23,37]. In fact, a negative DxWBS in patients with high suspicion of disease and/or increased Tg levels should not be reason to withhold I-131 therapy [24]. Still, any positive finding on DxWBS does provide an opportunity to fine-tune patient management. Although DxWBS diagnoses mere 25% of distant metastasis, it reliably detects thyroid remnants and lymph node metastasis [23,24,37,38,39]. Iodine-124 PET/CT is investigated as an alternative to I-123 DxWBS. Although disappointing diagnostic accuracy was reported previously, recent evidence shows that better results are feasible through diagnostic optimization using state-of-the-art PET/CT scanners [40,41].

In addition to potential discrepancies between DxWBS and TxWBS, insuperable inaccuracies in TxWBS assessments could have resulted in some false-positive readings. Any non-physiological uptake within the neck, but outside the thyroid bed, was ascribed to suspected lymph node metastasis. The a-priori chance that uptake foci read as metastases were true metastases, is most likely higher in patients with prior histopathology-proven lymph node metastasis (pN1). This is a limitation of the current study and of the interpretation of imaging studies in general.

Moreover, from the current retrospective study, it remains hypothetical whether pre-ablative knowledge of clinically relevant TxWBS findings would have actually influenced patient management, cumulative I-131 doses, and—more importantly—treatment success rate. To date no studies investigated the impact of DxWBS on long-term DTC treatment outcomes [9,21].

It is unknown whether additional costs for I-123 DxWBS (including altered I-131 doses, adjusted hospital admission durations, and additional diagnostic or surgical procedures) would outweigh the current costs for additional I-131 administrations and extended long-term follow-up following unsuccessful initial treatment. No formal cost-effectiveness studies exist. Local costs for a diagnostic 185 MBq dose of I-123 are €525 ($635, €1 = $1.21 on 26 February 2021) compared to €545–775 ($659–938) for any therapeutic dose of I-131. DxWBS using I-131 costs less (i.e., €88 or $106 for 75 MBq), but the radiation exposure is suggested to trigger targeted tissues to upregulate their cellular defense mechanisms. This could cause stunning: decreased sensitivity to the subsequent therapeutic I-131 dose when it is not delivered shortly after DxWBS. Although regularly disputed, it is still generally assumed that stunning results in less successful treatment. Stunning after I-131 DxWBS is well understood, but infrequently reported following I-123 DxWBS [11,42,43].

The current study has additional limitations. Its retrospective nature inevitably caused missing data, data heterogeneity, selection and information bias, and limits its level of evidence. Observer bias is likely for both qualitative and semi-quantitative measurements. Moreover, thyroid remnant size was quantified using a thyroid-to-background uptake ratio, in absence of a true quantitative method such as standardized acquisition protocols or normalization using a fixed-dose source scanned with the patient [44]. Also, although the definition of treatment success adhered to the current ATA guidelines, it ignored the patient category with solely a sTg of 0.5–1.0 ng/mL after nine months and no structural evidence of disease. These patients have an uncertain treatment outcome at first, but nearly always prove free of disease on successive follow-up without any subsequent I-131 treatments. Unfortunately, our study sample size was too small to distinguish this third outcome group separately. Study power could also have limited other observations, such as the operator dependency among individual surgeons.

Global controversy exists on the indication and optimal dose for I-131 therapy, mainly due to varying interpretation of the available literature and the lack of high-quality, decisive evidence on long-term outcomes of lower or higher dose I-131 therapy [20,45,46,47]. Due to the indolent nature of DTC, this requires large and preferably randomized-controlled cohort studies with a long follow-up—studies that are currently unavailable [20,45]. Any advocated recommendations for either higher or lower dose in any-risk DTC are currently more expert-based than evidence-based. Whereas European guidelines promote radioiodine in almost all DTC patients, more selective use of I-131 is advocated in the USA [14,17,18,19,44,48,49]. As such, the management consequences that are presumed in the current study if TxWBS findings had been diagnosed prior to I-131 therapy, may not reflect regular practice in other parts of the world. Whether lowering or more selective use of I-131 could safely be encouraged also depends on the local multidisciplinary expertise regarding DTC risk-assessment. Such considerations should be restricted to highly experienced and dedicated thyroid cancer centers [6].

## 5. Conclusions

In conclusion, additional pre-ablation diagnostics could contribute to improved risk stratification and treatment optimization in DTC patients referred for radioiodine treatment. We recommend selective application of DxWBS in patients with pN1 stage or patients at risk for in situ lymph node metastases. Treatment optimization based on pre-ablative knowledge of thyroid remnant size could enable accurate Tg-based follow-up, but seems unrelated to treatment success. Dependent on the institutional surgical expertise, we do not recommend routine DxWBS to determine thyroid remnant size. To provide stronger recommendations, cost-effectiveness of DxWBS and consequences of DxWBS-based management on improving the long-term oncological prognosis have yet to be investigated in future studies.

## Figures and Tables

**Figure 1 diagnostics-11-00553-f001:**
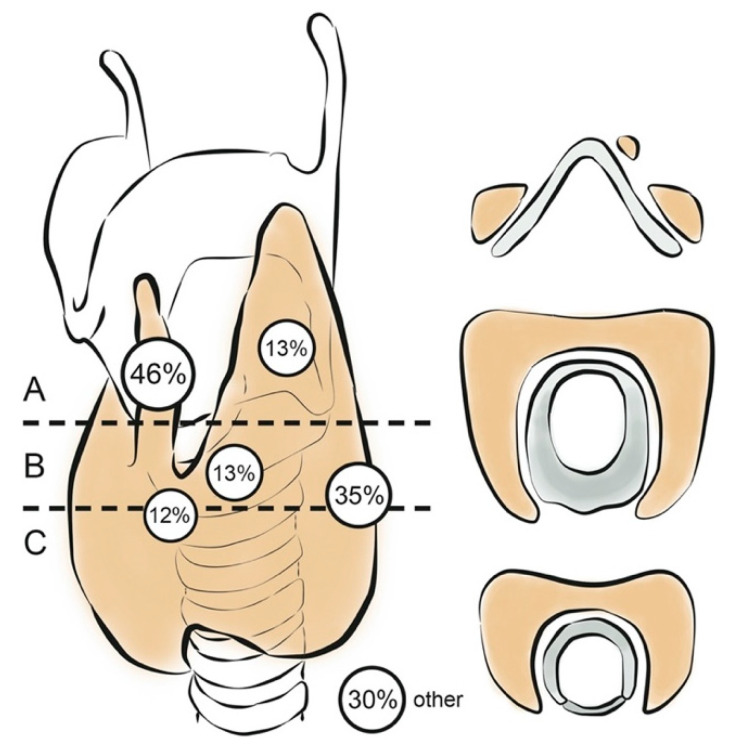
Thyroid remnant anatomical sites. Frequency of thyroid remnants at various anatomical sites within the thyroid bed. Forty-five (46%) remnants were found at the former pyramidal lobe, 34 (35%) at the tracheoesophageal groove, 13 (13%) at the former superior poles, 13 (13%) around Berry’s ligament, 12 (12%) at the former isthmus, and 29 (30%) in other locations, including the thymus and thyroglossal duct.

**Figure 2 diagnostics-11-00553-f002:**
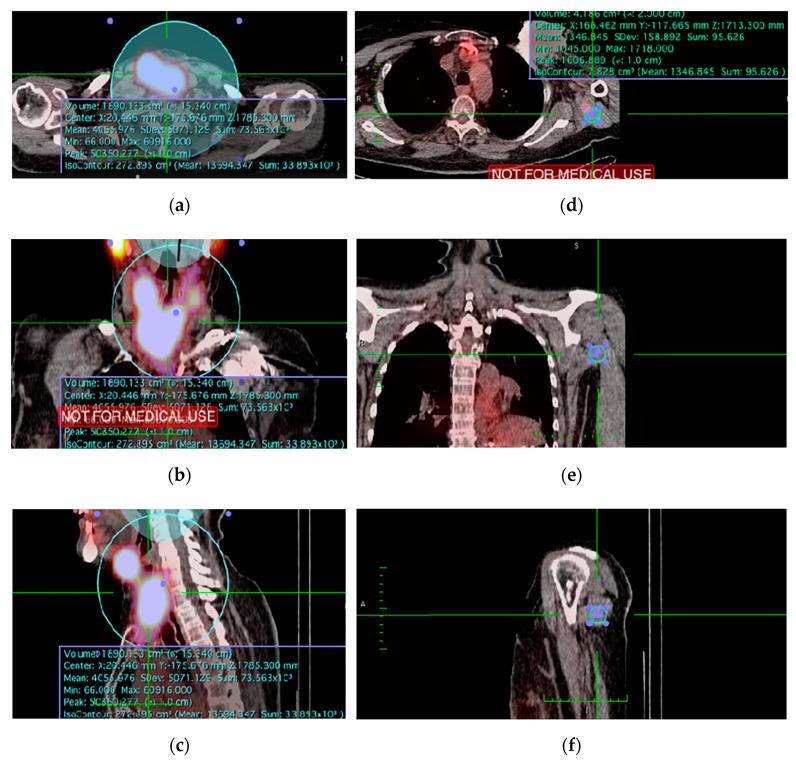
Example measurement of the thyroid remnant and shoulder volume-of-interest (VOI) on SPECT/CT: (**a**–**c**) Transverse, coronal and sagittal image of the semi-automatically drawn isocontour-based thyroid remnant VOI; (**d**–**f**) Transverse, coronal, and sagittal image of the predefined fixed-volume spherical background VOI (ø 20 mm) in the shoulder.

**Table 1 diagnostics-11-00553-t001:** Baseline demographics.

			*n* (%)
Total number of patients			97 (100%)
Age (years) (median, range)			45 (14–88)
Female			68 (70%)
Body mass index (kg/m^2^, IQR)			25.4 (22.3–30.6)
Ultrasound suspicious lymph nodes upon presentation			42 (43%)
Cytology confirmed lymph node metastases upon presentation			25 (25%)
Surgical intent	(suspicion of) malignancy		81 (84%)
	benign	goiter	10 (10%)
		parathyroidectomy	2 (2%)
		other	4 (4%)
Initial total thyroidectomy			68 (70%)
Two-stage surgery			29 (30%)
Lymph node dissection	formal neck dissection of ≥1 level		39 (40%)
	other		6 (6%)
Hospital of surgery	all surgeries at community hospital		6 (6%)
	all surgeries at tertiary care center		83 (86%)
	HT community, TT tertiary care		8 (8%)
Histopathology	Tumor size (mm, IQR)		23.0 (13.5–36.5)
	PTC	classic type	52 (54%)
		tall cell variant	2 (2%)
		diffuse sclerosing variant	7 (7%)
		oncocytic variant	1 (1%)
		cribriform variant	1 (1%)
	FVPTC		15 (15%)
	FTC		13 (13%)
	Hürthle cell carcinoma		3 (3%)
	poorly differentiated carcinoma		3 (3%)
	lymph node metastasis		43 (44%)
ATA risk	low		18 (19%)
	intermediate		38 (39%)
	high		41 (42%)
I-131 preparation	thyroid hormone withdrawal		91 (94%)
	rhTSH		6 (6%)

FTC: follicular thyroid carcinoma. FVPTC: follicular variant PTC. IQR: interquartile range. PTC: papillary thyroid carcinoma. rhTSH: recombinant TSH.

**Table 2 diagnostics-11-00553-t002:** Factors associated with thyroid remnant size (*n* = 87).

		TRB Ratio	
	N	Median (IQR)	*p*-Value
Overall thyroid remnant size	87	11.6 (7.07–28.7)	
**Clinical factors**			
Age (years)	87	R^2^ = 0.142 ^a^	0.190 ^a^
Female sex	58	10.85 (5.67–27.3)	0.331 ^b^
Body mass index (kg/m^2^)	87	R^2^ = 0.012 ^a^	0.914 ^a^
History of neck surgery	7	6.20 (1.86–57.56)	0.318 ^b^
**Peri-operative factors**			
Surgery with oncological intent	73	10.78 (5.13–24.44)	0.009 ^b^
Two-stage thyroidectomy	25	16.05 (9.52–76.70)	0.013 ^b^
Lymph node dissection			
no	51	16.05 (9.02–34.70)	0.003 ^c^
central compartment (level 6/7)	7	1.86 (0.46–8.13)	
central + lateral compartment (level 2–7)	29	9.55 (4.76–22.48)	
Leading surgeon			
surgeon A	44	9.60 (4.67–21.22)	0.136 ^c^
surgeon B	25	16.22 (9.88–47.37)	
surgeon C	7	19.37 (6.20–144.17)	
other	11	12.36 (7.07–18.75)	
Duo of dedicated thyroid surgeons	33	9.65 (4.51–23.53)	0.046 ^b^
≥1 surgery at referring hospital	12	10.29 (8.51–54.46)	0.410 ^b^
Difficult surgical procedure *	36	13.14 (7.80–27.92)	0.263 ^b^
Recurrent laryngeal nerve injury	9	10.92 (8.52–50.72)	0.540 ^b^
**Factors related to I-131**			
Dose I-131			
1.1 GBq	4	121.47 (36.73–366.66)	0.061 ^c^
3.7 GBq	31	12.36 (7.28–29.82)	
5.55 GBq	46	9.89 (5.31–27.36)	
7.4 GBq	6	14.64 (3.59–19.42)	
Preparation with rhTSH	6	17.87 (10.12–126.84)	0.150 ^b^

IQR: interquartile range. TRB ratio: thyroid-remnant-to-background ratio. rhTSH: recombinant TSH. *: difficult surgical procedure included: tissue adhesions, scar tissue, aberrant course of the recurrent laryngeal nerve, difficult access to retrotracheal thyroid remnant and other anatomic variations. ^a^: Spearman’s correlation coefficient. ^b^: Mann–Whitney U test. ^c^: Kruskal–Wallis test.

**Table 3 diagnostics-11-00553-t003:** Factors associated with treatment success (*n* = 97).

	Successful Treatment (*n* = 41)	Unsuccessful Treatment (*n* = 56)	*p*-Value
**Clinical factors**			
Age (years, median (IQR))	43.8 (38.8–52.9)	49.0 (35.1–65.6)	0.179 ^d^
Female sex	27 (65.9%)	41 (73.2%)	0.503 ^e^
Body mass index (kg/m^2^)	25.4 (22.9–28.9)	25.5 (22.1–31.4)	0.826 ^d^
History of neck surgery	3 (7.3%)	4 (7.1%)	1.000 ^f^
**Peri-operative factors**			
Surgery with oncological intent	32 (78.0%)	49 (87.5%)	0.215 ^e^
Two-stage surgery	15 (36.6%)	14 (25.0%)	0.264 ^e^
Lymph node dissection			
no	31 (75.6%)	27 (48.2%)	0.022 ^e^
central compartment (level 6/7)	2 (4.9%)	5 (8.9%)	
central + lateral compartment (level 2–7)	8 (19.5%)	24 (42.9%)	
Leading surgeon			
surgeon A	19 (46.3%)	27 (48.2%)	0.594 ^e^
surgeon B	12 (29.3%)	16 (28.6%)	
surgeon C	6 (14.6%)	4 (7.1%)	
other	4 (9.8%)	9 (16.1%)	
Duo of dedicated thyroid surgeons	9 (34.6%)	16 (38.1%)	0.802 ^e^
≥1 surgery at referring hospital	2 (4.9%)	12 (21.4%)	0.038 ^e^
Difficult surgical procedure *	10 (24.4%)	28% (50.0%)	0.012 ^e^
**Histopathological factors**			
Histopathology			
PTC	25 (61.0%)	38 (67.9%)	0.458 ^e^
FVPTC	7 (17.1%)	8 (14.3%)	
FTC	7 (17.1%)	6 (10.7%)	
Hürthle cell carcinoma	2 (4.9%)	1 (1.8%)	
poorly differentiated carcinoma	0 (0.0%)	3 (5.4%)	
Tumour characteristics			
tumour size (mm, median (IQR))	25.0 (14.5–30.0)	22.0 (13.0–41.0)	0.544 ^b^
aggressive tumour type **	3 (7.3%)	13 (23.2%)	0.052 ^f^
tumour multifocality	16 (39.0%)	23 (41.1%)	1.000 ^e^
capsular invasion			
Minimally invasive	5 (12.2%)	4 (7.1%)	0.415 ^f^
Invasive	8 (19.5%)	7 (12.5%)	
vascular invasion	13 (31.7%)	19 (33.9%)	0.831 ^e^
extrathyroidal extension	12 (29.3%)	26 (46.4%)	0.097 ^e^
irradical resection (R1) ***	10 (24.4%)	27 (48.2%)	0.021 ^e^
ATA risk classification			
low risk	11 (26.8%)	7 (12.5%)	0.157 ^e^
intermediate risk	16 (39.0%)	22 (39.3%)	
high risk	14 (34.1%)	27 (48.2%)	
pTNM classification			
pT1a	5 (12.2%)	4 (7.1%)	0.079 ^e^
pT1b	7 (17.1%)	11 (19.6%)	
pT2	14 (34.1%)	9 (16.1%)	
pT3	15 (36.6%)	27 (48.2%)	
pT4a	0 (0.0%)	5 (8.9%)	
pT4b	n.a.	n.a.	
Lymph node metastases			
Nx	10 (24.4%)	5 (8.9%)	0.030 ^e^
N0	19 (46.3%)	19 (33.9%)	
N1a	4 (9.8%)	8 (14.3%)	
N1b	8 (19.5%)	24 (42.9%)	
**Factors related to I-131 therapy**			
Administered Dose I-131			
1.1 GBq	3 (7.3%)	1 (1.8%)	0.005 ^e^
3.7 GBq	21 (51.2%)	15 (26.8%)	
5.55 GBq	17 (41.5%)	34 (60.7%)	
7.4 GBq	0 (0%)	6 (10.7%)	
Thyroid remnant size (median TRB ratio, IQR) (*n* = 87)	15.92 (8.48–44.37)	9.89 (5.31–22.10)	0.039 ^b^
Clinically relevant findings on TxWBS			
large remnant (TRB ratio ≥24.4)	16 (40.5%)	11 (20.0%)	0.042 ^e^
lymph node metastasis (*n* = 26)	7 (17.1%)	19 (33.9%)	0.103 ^f^
lymph node metastasis >10mm (*n* = 4)	0 (0%)	4 (7.1%)	0.135 ^f^
distant metastasis (*n* = 6)	0 (0%)	6 (10.7%)	0.037 ^f^
sTg at time of I-131 administration			
<0.5 ng/mL	15 (36.6%)	11 (20.4%)	0.033 ^e^
0.5–1.0 ng/mL	5 (12.2%)	2 (3.7%)	
>1.0 ng/mL	21 (51.2%)	41 (75.9%)	

FTC: follicular thyroid carcinoma. FVPTC: follicular variant PTC. IQR: interquartile range. n.a.: not applicable. PTC: papillary thyroid carcinoma. TRB ratio: remnant-to-background ratio. sTg: stimulated Tg. TxWBS: post-therapy I-131 whole-body scintigraphy. *: difficult surgical procedure included: tissue adhesions, scar tissue, aberrant course of the recurrent laryngeal nerve, difficult access to retrotracheal thyroid remnant and other anatomic variations. **: i.e., diffuse sclerosing, tall cell or oncocytic variant of PTC, Hürthle cell carcinoma, and poorly differentiated thyroid carcinoma. ***: Irradical resection (R1): if the carcinoma extended into less than one millimeter from the surgical plane [31]. ^b^: Mann–Whitney U test. ^d^: *t*-test. ^e^: Pearson chi-square exact test. ^f^: Fisher’s Exact test.

**Table 4 diagnostics-11-00553-t004:** Multivariable logistic regression model showing independent predictors of a successful treatment at nine months.

	Significance	Odds Ratio (95% CI)
Thyroid surgery at tertiary care center	0.022	7.094 (1.321–38.108)
Aggressive histopathological tumor type	0.036	0.168 (0.032–0.892)
TNM N1b stage	0.013	0.261 (0.091–0.751)
New lymph node metastasis on TxWBS	0.046	0.327 (0.109–0.981)
*Constant*	*0.188*	*0.346*

## Data Availability

The data presented in this study are available on request from the corresponding author. The data are not publicly available due to privacy regulations.

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
