# Peer review of "Radioiodine in Differentiated Thyroid Carcinoma: Do We Need Diagnostic Pre-Ablation Iodine-123 Scintigraphy to Optimize Treatment?"

_diagnostics, 2021, doi:10.3390/diagnostics11030553_

Round 1
Reviewer 1 Report
minor revisions such as improvement in bibliography ( latest papers published), text editing and improvement in originality.
Author Response
We would like to thank reviewer 1 for reviewing our manuscript.
Conform the advice, we have updated the bibliography with some recent papers on the subject. We have also reviewed and improved the English language and style, where appropriate.
Reviewer 2 Report
I enjoyed very much reading this manuscript that reflect real-life experiences from patient treatment. I commend the authors on having taken a balanced approach to the issue as opposed to the polemic that is often encountered between different practices, schools, disciplines, etc. involved in the treatment of differentiated thyroid carcinoma. Overall, the manuscript has very good utility for practicing physicians. There are still some points that in my opinion can be improved:
- To avoid confusion, the fact that the sensitivity of a diagnostic scan may be lower than that of a post-therapy scan should be mentioned already in the Introduction.
- It should be explained in the text the study was limited to patients treated between 2010 and 2015. One would normally expect to include patients treated as recently as 2019 or 2020, so this raises some questions.
- In lines 192-194, I can understand why the presence of metastasis on TxWBS is not included in the multi-variate analysis because it is “explicitly related to the defined outcome”, but I don’t see how this applies to T-stage or I-131 activity. The authors should explain this better in the text.
- In line 199, the authors state that only 2 (out of 99) patients underwent DxWBS, meaning that they themselves do not follow this practice (at least in 2010-2015). This should be stated more explicitly.
- It seems counter-intuitive that a smaller thyroid remnant would be associated with unsuccessful treatment. This is likely not a direct association (especially since it was not a significant predictor in the multivariate analysis), but it still warrants some discussion (e.g., these patients were maybe operated by more experienced surgeon teams because they had metastatic disease, etc.).
- Lines 299-301: Can the authors propose a plausible biological interpretation for this counter-intuitive observation?
- Line 306 should read « would have likely been altered »
- Line 311 should read “would have also been likely”
- Lines 307-301: It should stated here that among the 27 patients with large thyroid remnants, 16 had a sucessful treatment and 11 had an unsuccessful treatment. It should also be reminded that a large thyroid remnant was not a predictive factor for success. Therefore, saying here that these patients would have received a higher I-131 activity does not seem appropriate and justified by the manuscript’s data. This is discussed in the Discussion section, but it should also be mentioned here, because it is quite striking for the reader.
- Lines 311-314: It should be stated that a higher activity in case of lymph node metastasis reflects the author’s local (or national) practice, and would not necessarily apply in other centers or other countries.
- Line 311: 26 of the 97 patients were found to have a suspicion of metastatic lymph nodes on the TxWBS. This percentage (>25%) seems quite high, and suggests that the pre-operative staging of these patients could most certainly be improved via better application of cervical ultrasound. This is very important, because better ultrasound skills can definitely improve patient outcomes, avoid reoperations, avoid patient anxiety related to residual disease and need for further treatments, etc. It is a much more tangible measure to improve outcomes than the (in large part academic) discussion about utility of DxWBS.
- Lines 350-352 mention the importance of appropriate identification of metastatic lymph nodes, and the critical role of pre-operative ultrasound should be stressed here as well.
- It is also striking that there is no mention in the manuscript regarding whether the potential metastatic lymph nodes were verified by ultrasound or not. This should be mentioned in conjunction with the limitation discussed in line 387.
- Lines 335-337: This sentence should go to the beginning of the paragraph, not the end, because otherwise it may be taken as a negation of what has come before.
- Lines 345-359: Here again, it should be acknowledged that these practices vary widely among centers and countries.
- Line 377: Again, the manuscript’s data show that remnant size per se does not matter. Regarding the metastatic lymph nodes, it is much better to see them before surgery rather after I-131, and here is where ultrasound expertise is key.
Author Response
We would like to thank reviewer 2 for the thorough review of our manuscript.
Please see below for a point-by-point reply to the provided feedback.
- To avoid confusion, the fact that the sensitivity of a diagnostic scan may be lower than that of a post-therapy scan should be mentioned already in the Introduction.
We nuanced this in line 69-70 of the introduction.
- It should be explained in the text the study was limited to patients treated between 2010 and 2015. One would normally expect to include patients treated as recently as 2019 or 2020, so this raises some questions.
Conceptualization and data extraction for the current study started as early as 2019. In 2015, new Dutch guidelines for the treatment of thyroid carcinoma were introduced. We did not want heterogeneity from two different sets of guidelines to influence our analysis. Moreover, we needed a sufficient number of eligible patients and at least 9-12 months of available follow-up from all included patients, to include their first follow-up moment. This is why, back in 2019, we decided to study the patients from before 2015 only.
- In lines 192-194, I can understand why the presence of metastasis on TxWBS is not included in the multi-variate analysis because it is “explicitly related to the defined outcome”, but I don’t see how this applies to T-stage or I-131 activity. The authors should explain this better in the text.
Thank you for pointing out this mistake in the text. Presence of metastasis of TxWBS was a constant and therefore not included in the multivariate analysis. T-stage and I-131 activity showed multicollinearity (p<0.001, Pearson’s chi square) (as did N-stage and I-131 activity, p<0.001), but are not a constant in relation to the outcome.
The multivariable logistic regression was repeated including T-stage. I-131 activity was excluded due to the observed multicollinearity. There were no major changes to the results of the regression. Minor numerical changes to the odds ratios and their confidence intervals were adjusted in Table 4 and the corresponding text. Explanations of the excluded independent variables were moved to the results section, as appropriate.
- In line 199, the authors state that only 2 (out of 99) patients underwent DxWBS, meaning that they themselves do not follow this practice (at least in 2010-2015). This should be stated more explicitly.
This is stated in line 72-75 of the introduction.
- It seems counter-intuitive that a smaller thyroid remnant would be associated with unsuccessful treatment. This is likely not a direct association (especially since it was not a significant predictor in the multivariate analysis), but it still warrants some discussion (e.g., these patients were maybe operated by more experienced surgeon teams because they had metastatic disease, etc.).
Our hypothesis was similar to that of the reviewer. Table 2 shows that thyroid remnant size was related to the experience of the thyroid surgeons: a smaller remnant was seen if a team of experienced, dedicated surgeons operated. A smaller remnant is also seen when lymph node dissection was performed.
More experienced surgeons did operate the more complex cases, such as patients with lymph node metastasis (p=0.003). It is likely that they operated these cases with greater precision, leaving a smaller thyroid remnant. This may case the unexpected inverse association between a small remnant and treatment success. As multivariate analysis showed that thyroid remnant size was not independently associated with treatment success, but pN1 stage was, we chose not to elaborate on this in the (already lengthy) discussion. Upon request of reviewer 2, we have now added it to the discussion (line 338-347).
- Lines 299-301: Can the authors propose a plausible biological interpretation for this counter-intuitive observation?
We believe that any existing association between suspected lymph node metastasis on TxWBS and unsuccessful treatment in pN0/x patients may be minor and mainly limited by study power. Unknown confounding variables may exist (as briefly explained in Discussion line 384-386).
One additional explanation, which we cannot support by data from our study, is that observations of suspected lymph node metastasis on TxWBS may include a small number of false-positive readings. We speculate that in patients with pN1 stage, the chance of a false positive reading is smaller than in pN0/x patients. This was also discussed in the study limitations (Discussion, line 406-409)
- Line 306 should read « would have likely been altered »
We have processed this suggestion.
- Line 311 should read “would have also been likely”
We have processed this suggestion.
- Lines 307-301: It should stated here that among the 27 patients with large thyroid remnants, 16 had a sucessful treatment and 11 had an unsuccessful treatment. It should also be reminded that a large thyroid remnant was not a predictive factor for success. Therefore, saying here that these patients would have received a higher I-131 activity does not seem appropriate and justified by the manuscript’s data. This is discussed in the Discussion section, but it should also be mentioned here, because it is quite striking for the reader.
Paragraph 3.5 describes the management changes in these patients in light of the Dutch national guidelines. Whether such changes would be justified considering the findings of our study, was not assessed. After all, we cannot be certain of any potential impact of these management changes on the 9-month treatment success rate. Following the reviewers advice, we did add to this paragraph (line 309-310) that a higher dose for a larger thyroid remnant may not be justified.
- Lines 311-314: It should be stated that a higher activity in case of lymph node metastasis reflects the author’s local (or national) practice, and would not necessarily apply in other centers or other countries.
It is true that these presumed management changes merely reflect our national guidelines. As there are major global variations in the administration of radioiodine, these changes do not necessarily reflect the practice in other countries. We have specified this in a more encompassing remark in the corresponding paragraph at the end of the discussion (line 446-450).
- Line 311: 26 of the 97 patients were found to have a suspicion of metastatic lymph nodes on the TxWBS. This percentage (>25%) seems quite high, and suggests that the pre-operative staging of these patients could most certainly be improved via better application of cervical ultrasound. This is very important, because better ultrasound skills can definitely improve patient outcomes, avoid reoperations, avoid patient anxiety related to residual disease and need for further treatments, etc. It is a much more tangible measure to improve outcomes than the (in large part academic) discussion about utility of DxWBS.
We agree with the reviewer that optimization of preoperative ultrasound procedures is crucial, and more important than the introduction of a more laborious technique. Complete cervical ultrasound exams and reports will aid multidisciplinary discussions on the most suitable course of treatment and in the end improve individual patient outcomes. At our institution, dedicated radiologists are already involved in these procedures. Suspected lymph node metastasis on TxWBS in the current study were mostly non-enlarged iodine-positive nodes. It is uncertain how many of those would have been detected on a detailed ultrasound exam. Moreover, this rate of newly diagnosed lymph node metastasis on TxWBS corresponds to previously published numbers.
We have clarified this in line 362-370 of the discussion.
- Lines 350-352 mention the importance of appropriate identification of metastatic lymph nodes, and the critical role of pre-operative ultrasound should be stressed here as well.
Please see our reply to comment 11.
- It is also striking that there is no mention in the manuscript regarding whether the potential metastatic lymph nodes were verified by ultrasound or not. This should be mentioned in conjunction with the limitation discussed in line 387.
At our hospital, suspected lymph node metastasis diagnosed on TxWBS are first re-evaluated at the 4-6 months follow-up moment after TxWBS, using ultrasound (+FNAC) and unstimulated Tg (sTg, and DxWBS upon indication will follow approximately 9 months after radioiodine treatment). In the current study, suspected lymph node metastasis were no longer visualized in 20 of 26 (77%) of patients. In the remaining 6 patients, all six were confirmed on DxWBS at 9 months. Three of them were also confirmed on ultrasound. We have clarified this in this paragraph of the discussion (line 371-373). In general, our surgeons only proceed to cervical lymph node resection in case nodes are larger than 10mm in diameter and proven malignant (FNA or iodine-avid).
- Lines 335-337: This sentence should go to the beginning of the paragraph, not the end, because otherwise it may be taken as a negation of what has come before.
Following the reviewers advice, we have reformulated this paragraph.
- Lines 345-359: Here again, it should be acknowledged that these practices vary widely among centers and countries.
To these lines, we have added the phrase ‘according to our local protocol’.
Please also see our reply to comment 10.
- Line 377: Again, the manuscript’s data show that remnant size per se does not matter. Regarding the metastatic lymph nodes, it is much better to see them before surgery rather after I-131, and here is where ultrasound expertise is key.
Please see our reply to comment 11.

Round 2
Reviewer 2 Report
All my comments have been fully well addressed.